# Heat Shock Factor 1 Directly Regulates Postsynaptic Scaffolding PSD-95 in Aging and Huntington’s Disease and Influences Striatal Synaptic Density

**DOI:** 10.3390/ijms222313113

**Published:** 2021-12-04

**Authors:** Nicole Zarate, Taylor A. Intihar, Dahyun Yu, Jacob Sawyer, Wei Tsai, Maha Syed, Luke Carlson, Rocio Gomez-Pastor

**Affiliations:** Department of Neuroscience, School of Medicine, University of Minnesota, Minneapolis, MN 55455, USA; zarat013@umn.edu (N.Z.); intih006@umn.edu (T.A.I.); Yu.Dahyun@mayo.edu (D.Y.); sawye271@umn.edu (J.S.); tsaix246@umn.edu (W.T.); syed0022@umn.edu (M.S.); carl5146@umn.edu (L.C.)

**Keywords:** HSF1, PSD-95, aging, Huntington’s disease

## Abstract

PSD-95 (*Dlg4*) is an ionotropic glutamate receptor scaffolding protein essential in synapse stability and neurotransmission. PSD-95 levels are reduced during aging and in neurodegenerative diseases like Huntington’s disease (HD), and it is believed to contribute to synaptic dysfunction and behavioral deficits. However, the mechanism responsible for PSD-95 dysregulation under these conditions is unknown. The Heat Shock transcription Factor 1 (HSF1), canonically known for its role in protein homeostasis, is also depleted in both aging and HD. Synaptic protein levels, including PSD-95, are influenced by alterations in HSF1 levels and activity, but the direct regulatory relationship between PSD-95 and HSF1 has yet to be determined. Here, we showed that HSF1 chronic or acute reduction in cell lines and mice decreased PSD-95 expression. Furthermore, *Hsf1*^(+/−)^ mice had reduced PSD-95 synaptic puncta that paralleled a loss in thalamo-striatal excitatory synapses, an important circuit disrupted early in HD. We demonstrated that HSF1 binds to regulatory elements present in the PSD-95 gene and directly regulates PSD-95 expression. HSF1 DNA-binding on the PSD-95 gene was disrupted in an age-dependent manner in WT mice and worsened in HD cells and mice, leading to reduced PSD-95 levels. These results demonstrate a direct role of HSF1 in synaptic gene regulation that has important implications in synapse maintenance in basal and pathological conditions.

## 1. Introduction

The postsynaptic scaffolding protein PSD-95 (*Dlg4*) is a member of the membrane-associated guanylate kinase (MAGUK) family of proteins known for anchoring ionotropic glutamate receptors (AMPAR and NMDARs) to the membrane. Changes in the levels of PSD-95 alter clustering and maintenance of glutamate receptors, thus playing an essential role in regulating synaptic transmission and plasticity [1,2]. Glutamatergic synaptic transmission and plasticity are fundamental mechanisms contributing to memory and cognition. Transcriptional dysregulation of PSD-95 has been reported during aging and in several neurodegenerative diseases (NDs) including Alzheimer’s (AD) and Huntington’s disease (HD). Depletion of PSD-95 is believed to contribute to alterations in synaptic function and behavioral deficits [3,4,5,6], but the mechanisms involved in the dysregulation of PSD-95 under these pathological conditions are not fully understood. Previous studies have identified several transcriptional regulators of *Dlg4* including the Ikaros family zinc finger transcription factor Eos and Early Growth Response 1 (Egr-1) [7,8,9]. Our understanding of the transcriptional regulation of PSD-95 primarily relies on neurodevelopmental studies where these regulators are expressed in abundance. However, the expressions of some of these regulators decline in the adult brain [7,8,10]. Therefore, alternative unknown mechanisms for the regulation of PSD-95 might exist in the fully developed brain.

The Heat Shock Transcription Factor 1 (HSF1), traditionally known for its role in regulating stress response and protein homeostasis [11], was recently proposed as a regulator of synapse stability and memory consolidation [5,11,12,13,14]. This is supported by studies showing basal as well as stress-dependent accumulation of various chaperones within synaptosomes (isolated synaptic terminals) where they participate in modulating synaptic protein homeostasis [15,16]. Recent studies suggested a more direct role of HSF1 in the regulation of various synaptic components. In line with these studies, *Hsf1*^KO^ mice showed aberrant synapse formation in the hippocampus, altered expression of synaptic proteins like the polysialylated-neural cell adhesion molecule (PSA-NCAM) and PSD-95, and working memory deficits as well as other behavioral alterations [14,17,18]. Additional genetic and pharmacological manipulations aimed at activating HSF1 under both basal and pathological conditions resulted in increased levels of select synaptic proteins including PSD-95, Synapsin I, and Synaptophysin (SYP1) [5,17]. Therefore, it is reasonable to hypothesize that HSF1 plays a direct role in the regulation of synaptic function by directly controlling the expression of specific synaptic genes.

HSF1 protein levels decline during aging, a phenomenon that is exacerbated in different NDs like HD where HSF1 is reported to be abnormally degraded [19,20,21,22,23]. HD is a fatal neurodegenerative disorder manifested by motor and cognitive decline. HD is caused by a polyglutamine repeat expansion in the *Htt* gene, resulting in a mutant form of the HTT protein (mtHTT) prone to misfolding and aggregation [24] that preferentially affects neurons of the striatum. Down-regulation of PSD-95 within the striatum is considered a pathological marker in HD and reflects synaptic dysfunction [25,26], but, whether reduction of HSF1 in aging or HD directly contributes to PSD-95 dysregulation and synaptic dysfunction, has yet to be determined.

In this study, we sought to determine whether HSF1 is a direct regulator of PSD-95 expression. We focused our analyses on both immortalized striatal cells and striatum tissues given the role of this brain region in the regulation of behavioral deficits in both aging and HD, and in which changes of PSD-95 and HSF1 were previously reported [27,28,29,30,31,32]. We showed that PSD-95 and HSF1 protein levels decrease in parallel in the striatum of WT mice in an age-dependent manner, a phenomenon that is exacerbated in the zQ175 HD mouse model. We established proof of concept for the role of HSF1 in regulating PSD-95 expression and synapse stability by conducting chronic and acute reduction of HSF1 in striatal cells and mice, which resulted in the downregulation of PSD-95 and loss of striatal excitatory synapse density. Finally, we demonstrated that HSF1 binds to regulatory elements (Heat Shock Elements, HSEs) present in both murine and human *Dlg4* genes, and that such binding decreased over time and in the presence of mtHTT, coinciding with transcriptional reduction of PSD-95. Overall, our data provides strong evidence for the regulatory role of HSF1 on PSD-95 expression and highlights the importance of this regulatory interaction in the maintenance of striatal glutamatergic synapses in both physiology and disease.

## 2. Results

### 2.1. Aging-Related Reduction of PSD-95 and HSF1 Is Increased in HD

Previous studies showed that HSF1 and PSD-95 decreased in an age-dependent manner in mouse models as well as humans [4,11,33,34,35], although the relationship between the alterations in those proteins has not yet been established. We therefore investigated whether age-dependent dysregulation of HSF1 relates to PSD-95 reduction. We performed immunoblot analyses of HSF1 and PSD-95 in striatal tissue from WT mice at 3, 6, 12, and 22 months old (Figure 1A,B). HSF1 protein levels significantly decreased between 3 and 6 months (~26% reduction, *p* = 0.023) and reached their lowest levels at 22 months (~46% reduction from 3 months, *p* = 0.0002). Similarly, PSD-95 levels decreased over time, with the lowest concentration at 22 months (~54% reduction from 3 months, *p* = 0.0008). These data recapitulate previous reports and indicate a parallel time-dependent reduction of these two proteins in the striatum of WT mice.

In HD, reduction of PSD-95 is particularly relevant for its role in dysregulation of striatal glutamatergic synapses. HD mouse models and postmortem tissue from patients with HD have shown reduced levels of PSD-95 in the striatum compared to control individuals [25,36,37,38]. We previously showed that HSF1 is abnormally degraded in HD [19]. Given our observation that HSF1 reduction parallels PSD-95 reduction in WT animals over time, we explored whether HSF1 and PSD-95 levels were concomitantly reduced in HD. We used the heterozygous zQ175 HD mouse model [30] and performed immunohistochemical and immunoblotting analyses for PSD-95 (Figure 1C–F). We examined the dorsolateral striatum of mice harvested at 5 weeks, a time point at which excitatory synapse deficits are already shown [19,26], as well as 6 months, a time point characterized by significant mtHTT aggregation, the onset of motor symptoms, and global transcriptional deficits [30,39] (Figure 1C). We observed that zQ175 PSD-95 intensity was significantly lower than in WT mice at 5 weeks (*p* = 0.0002) and as well as at 6 months (*p* = 0.00628). Comparisons within genotypes also showed a significant reduction of PSD-95 over time.

Immunoblot analyses of striatal tissue from 6 month WT and zQ175 mice confirmed a reduction in both HSF1 and PSD-95 in HD mice (Figure 1E,F). Reduction of HSF1 in HD has been previously connected to post-translational events [19], but downregulation of PSD-95 seems to be caused by transcriptional dysregulation. We confirmed that the reduction of PSD-95 protein levels is consistent at the transcript level by RT-qPCR analyses of *Dlg4* mRNA levels in zQ175 mice compared to WT, in an immortalized striatal cell model of HD (ST*Hdh*^Q111^) compared to control (ST*Hdh*^Q7^) and in postmortem striatal samples from patients with HD (Appendix A), as previously reported [6]. These data demonstrated that PSD-95 reduction is sustained in an age-dependent manner alongside HSF1, and when comparing WT and zQ175 mice at 6 months, the reduction in HSF1 and PSD-95 levels was exacerbated.

### 2.2. Acute or Chronic Reduction of HSF1 Results in Reduced PSD-95 Expression

We hypothesized that the reduction of HSF1 is responsible for the transcriptional dysregulation of *Dlg4* in both aging and HD and, therefore, sought to characterize how direct modification of HSF1 levels influences *Dlg4* expression. As proof of concept, ST*Hdh*^Q7^ cells were transfected with an siRNA targeting *Hsf1* (si*Hsf1*) followed by immunoblotting (Figure 2A). Cells transfected with non-targeting siRNA (Sscr) were used as a negative control. HSF1 knockdown resulted in a ~70% reduction in HSF1 protein levels (*p* = 0.002) and was accompanied by a ~50% reduction in PSD-95 protein levels (*p* = 0.017) (Figure 2A,B).

The use of siRNA showed that transient and acute reduction of HSF1 significantly impacted the levels of PSD-95. However, in both aging and neurodegeneration, reduction of HSF1 occurs in a sustained manner. To better mimic this chronic reduction of HSF1, we generated *Hsf1* heterozygous (+/−) cells for both ST*Hdh*^Q7^ and ST*Hdh*^Q111^ using the CRISPR/Cas9 system. The guide RNA (gRNA) was targeted to a region within exon 9 of the *Hsf1* gene, and single cell clones were isolated and genotyped. We selected clone 3A4 for ST*Hdh*^Q7^ and clone 8E6 for ST*Hdh*^Q111^ cells containing a 248 bp and 321 bp deletion, respectively, in one allele of *Hsf1* (Figure 2C,D). Immunoblotting confirmed a significant reduction in the levels of HSF1 in both ST*Hdh*^Q7^:*Hsf1*^(+/−)^ (*p* = 0.0056) and ST*Hdh*^Q111^:*Hsf1*^(+/−)^ (*p* = 0.0022) cells (Figure 2E,F). Chronic reduction of HSF1 in both ST*Hdh* cell lines showed a significant reduction in PSD-95 protein levels by 74.8% in Q7 and 30.6% in Q111 when compared with their corresponding control cell line (Figure 2G). Taken together, these data show that altering HSF1 levels by either acute and transient silencing with siRNA or by knocking-out one allele *of Hsf1* decrease of PSD-95 protein levels.

Due to the significant changes in PSD-95 levels when directly modulating HSF1 levels in vitro, we explored whether disruption of HSF1 in vivo would affect PSD-95 expression in a similar manner by using heterozygous *Hsf1*^(+/−)^ mice. Immunoblot analyses of HSF1 and Hsp70, a well-known canonical target of HSF1 [11], showed a reduction in these two proteins in *Hsf1*^(+/−)^ mice compared to WT (HSF1: *p* = 0.007, Hsp70: *p* = 0.012), confirming the negative impact of HSF1 haploinsufficiency in the expression of well-known HSF1 targets (Figure 3A,B).

PSD-95 immunofluorescence (IF) in the dorsolateral striatum of *Hsf1*^(+/−)^ mice revealed a significant reduction in the number of PSD-95 puncta (associated with synapses) compared to WT (*p* < 0.0001) (Figure 3C,D). Reduced PSD-95 protein levels were also confirmed in the cortex of *Hsf1*^(+/−)^ mice (Appendix A). We also tested the expression of SYP1, previously suggested to be regulated by HSF1 [5,13], but no changes were seen between genotypes (Appendix A). Previous studies in *Hsf1*^(−/−)^ mice showed reduced PSD-95, aberrant synapse formation, impaired spinogenesis in the hippocampus, and several behavioral alterations including reduced anxiety and working memory deficits [14,17,18]. Despite this evidence, this phenotype was associated with neurodevelopmental issues rather than a direct role of HSF1 in regulating PSD-95 and synapse stability.

Considering the role of PSD-95 in synapse formation and maintenance, we then assessed whether striatal synapse stability was affected in *Hsf1*^(+/−)^ mice. Ex vivo analyses of the dorsolateral striatum using colocalization of VGlut2, a marker for thalamic pre-synaptic input, and PSD-95 were used to quantify thalamo-striatal excitatory synapse density, a major synaptic circuit involved in striatal function [19,26,40,41]. Levels of thalamo-striatal synapses were reduced in *Hsf1*^(+/−)^ mice compared to WT (*p* < 0.0001), paralleling the reduction seen in PSD-95 puncta (Figure 3C,D), and indicating a worsening in synapse stability. Given these results as well as our previous in vitro studies, it can be concluded that modulating HSF1 levels leads to a reduction of PSD-95, an effect that can further lead to striatal synaptic dysfunction.

### 2.3. HSF1 Directly Binds to HSEs Present in the PSD-95 (Dlg4) Gene and Regulates Its Transcription

HSF1 regulates the expression of target genes by binding to HSEs, canonical sites that contain alternating inverted repeats of an nGAA sequence [11] (Figure 4A). We investigated whether there were HSEs within the promoter or intergenic regions of *Dlg4* that could explain a direct transcriptional regulation by HSF1. In silico analyses revealed four HSEs in the murine gene (m*Dlg4*) and one HSE in the human *DLG4* gene (h*DLG4*) (Figure 4B). Two HSEs (#1 and #2) were identified in the promoter region of the mouse *Dlg4* gene and two HSEs (#3 and #4) were located within the coding DNA region (Figure 4B). Interestingly, the only HSE identified in h*DLG4* (located in the promoter region) was identical to the murine HSE #3 (cTTCctGAA). HSF1 chromatin immunoprecipitation (HSF1-ChIP) on all four m*Dlg4* HSEs in ST*Hdh*^Q7^ cells showed a significant enrichment compared to IgG (negative control) at HSE #3, indicating HSF1 binds to this regulatory element in the murine *Dlg4* gene (Figure 4C). Therefore, we focused our subsequent HSF1-ChIP studies in brain samples on *Dlg4* HSE #3. Striatum samples were extracted from 3, 6, and 12 month WT animals to observe how HSF1 binding capacity to *Dlg4* is altered over time (Figure 4D). There were no significant differences in HSF1 binding to *Dlg4* between 3 and 6 months, although there was a significant reduction at 12 months (*p* = 0.013), indicating a significant age effect on HSF1 binding capacity. The timing at which we observed a significant reduction in HSF1 binding on *Dlg4* regulatory elements (between 6 and 12 months) preceded the reduction in protein levels of PSD-95 (between 12 and 22 months) (Figure 1A,B).

To determine whether HSF1 directly regulated the expression of PSD-95 we conducted luciferase assays with a reporter containing the h*DLG4* promoter, which had HSE #3 fused to luciferase (PSD95-pGL3a, obtained from Dr. Bao [8]), and we observed that luciferase activity was reduced after co-transfection with si*Hsf1*. (Figure 4E,F). We used site-directed mutagenesis to mutate the h*DLG4* HSE from GA->CT (Figure 4G), thereby ablating HSF1 binding [42], to determine if direct regulation of *Dlg4* expression by HSF1 is mediated through binding at HSE #3. ST*Hdh*^Q7^ cells transfected with the mutated h*DLG4* HSE (Mut. HSE) showed a significant reduction in luciferase activity compared with WT h*DLG4* HSE (Figure 4H). These results demonstrated that HSF1 directly binds to PSD-95 regulatory elements and regulates its transcription, supporting the hypothesis that changes in levels of HSF1 in the mouse brain can directly lead to decreased expression of PSD-95.

### 2.4. HSF1 Binding to Dlg4 Regulatory Elements Is Impaired in HD

Once established that HSF1 binds to and regulates *Dlg4* transcription, we assessed whether this regulatory mechanism was altered in HD where both HSF1 and PSD-95 protein levels and transcripts are reduced. HSF1-ChIP in ST*Hdh*^Q111^ cells compared to ST*Hdh*^Q7^ revealed that HSF1 binding to HSE #3 in *Dlg4* was significantly reduced (*p* = 0.011) (Figure 5A). Additionally, HSF1-ChIP studies in zQ175 mice showed that at 3 months (pre-symptomatic), there were no differences in HSF1 binding to HSE #3 in *Dlg4* (Figure 5B). At 6 months (symptomatic), HSF1 binding to HSE #3 was reduced (*p* = 0.0409) (Figure 5C), coinciding with the onset of HSF1 reduction in this mouse model [19]. HSF1-ChIP analyses in human striatal tissue confirmed HSF1 binding to *DLG4* HSE compared to IgG, although no significant changes were observed when comparing postmortem striatal tissue from unaffected individuals (control) and patients with HD (Figure 5D). Several factors may influence these results including (i) an insufficient number of samples to adequately addressed statistical significance, (ii) the lack of information regarding the cause of death, (iii) the mixed sex, and (iv) the difference in age and postmortem harvest time across all analyzed samples.

We conducted luciferase assays in ST*Hdh*^Q7^ and ST*Hdh*^Q111^ cells using the PSD95-pGL3a plasmid. The results obtained revealed a significant decrease in luciferase activity in HD cells relative to control cells (*p* < 0.0001) (Figure 5E). Furthermore, transfection of PSD95-pGL3a in ST*Hdh*^Q7^ and ST*Hdh*^Q111^ cells lacking one allele of *Hsf1* (ST*Hdh*^Q7^:*Hsf1*^(+/−)^ and ST*Hdh*^Q111^:*Hsf1*^(+/−)^) showed a significant reduction (~90%, *p* < 0.0001) in luciferase activity in ST*Hdh*^Q7^ when HSF1 levels were reduced (Figure 5F). Although our stringent statistical analysis did not reveal a significant difference in ST*Hdh*^Q111^:*Hsf1*^(+/−)^ vs ST*Hdh*^Q111^, we observed a trend towards a reduction in luciferase signal of ~50% in ST*Hdh*^Q111^:*Hsf1*^(+/−)^ (0.127 ± 0.06) compared to ST*Hdh*^Q111^ (0.265 ± 0.08). Taken together, our data indicated the pathological reduction of HSF1 in HD cell lines, and mouse models results in decreased HSF1 binding to PSD-95 and subsequent decreased expression of this synaptic protein.

## 3. Discussion

Age-related cognitive decline as well as motor and cognitive impairments seen in HD and other NDs are influenced by progressive synaptic dysfunction. However, the mechanisms responsible for synaptic dysregulation during both aging and neurodegeneration are still unclear [3,4,5,6,25,36,43]. In this study we have demonstrated that the stress protective transcription factor HSF1, known for its role in the regulation of protein quality control machinery and progressively depleted in aging and HD, contributes to the transcriptional regulation of the postsynaptic scaffolding gene *Dlg4*. These findings confirmed a regulatory role for HSF1 that has meaningful implications for synapse stability in aging and neurodegeneration.

The involvement of HSF1 in synapse regulation was previously associated with its role in regulating protein quality control systems and the expression of chaperones like HSP90, HSP60, and HSP70, which can be found in isolated synaptosomes from rat forebrain and cerebellum [15,16]. Chaperone levels decrease in aging and HD, which is influenced by the abnormal post-translational degradation of HSF1 under these conditions [11,19,44]. Therefore, it is reasonable to hypothesize that HSF1 degradation could impact the concentration and composition of the chaperone pool at synapses, thereby altering synaptic function. Alternatively, other studies have suggested a more direct role of HSF1 in the regulation of synaptic function by controlling synaptic components such as PSA-NCAM (polysialylated-neural cell adhesion molecule), known to participate in the remodeling of neuronal circuits [17]. In this study, the authors also reported reduced PSD-95 protein levels in the hippocampus of *Hsf1*^KO^ mice, which was reversed by the overexpression of a constitutively active form of HSF1 (caHSF1) in neonatal *Hsf1*^KO^ mice [17]. While the study did not assess how HSF1 participated in the regulation of PSD-95, it highlighted a potential connection between the levels of HSF1 and PSD-95.

Studies in AD mice or primary hippocampal neurons treated with 17-AAG, an HSP90 inhibitor that leads to activation of HSF1, resulted in increased expression of PSD-95 as well as the pre-synaptic proteins Synapsin I and SYP1, and BDNF (brain-derived neurotrophic factor) [5]. However, it was unclear whether the effects on the expression of these various synaptic components by 17-AAG were HSF1-dependent or if HSF1 was directly involved in their regulation. Ting and colleagues [45] showed that HSF1 directly binds to HSEs in the promoter of SAP97, another MAGUK family synaptic scaffolding protein with various functions in the regulation of synaptic receptor clustering, and regulated SAP97 expression in cardiomyocytes. We showed that manipulating the levels of HSF1 in vitro as well as in analyses in *Hsf1*^(+/−)^ mice leads to decreased expression and protein levels of PSD-95. We showed that the effect mediated on PSD-95 involved the direct binding of HSF1 onto an HSE within the *Dlg4* gene/promoter in both mice and humans. This was corroborated following the luciferase experiments showing that HSF1 directly regulates the expression of *Dlg4* in cells and mice. However, we did not observe changes in SYP1, implying that effects mediated by 17-AAG on these proteins could be mediated by factors other than HSF1.

HSF1 binding to *Dlg4* regulatory elements decreased during aging, which paralleled the reduction of HSF1 and PSD-95 protein levels. Intriguingly, we observed that HSF1 protein levels started to decrease between 3 and 6 months of age while HSF1 binding to *Dlg4* was significantly reduced between 6 and 12 months. PSD-95 protein was not significantly reduced until 12–22 months. This could indicate that reduction of HSF1 does not immediately translate to decreased binding. It is possible that HSF1 maintains its regulatory functions on some genes up to a certain threshold. Once HSF1 is reduced below a minimum functional concentration, the consequences of HSF1 reduction on DNA binding are more evident. While we showed a direct regulatory role of HSF1 on *Dlg4* expression, decreased HSF1 binding in vivo does not immediately translate into the reduction of PSD-95 protein levels. One possibility is that other unknown transcription factors, in conjunction with HSF1, contribute to maintaining some basal PSD-95 expression.

We showed that HSF1 binding on the *Dlg4* gene is reduced in both cell and mouse models of HD, which correlates with the pathological reduction of HSF1 and PSD-95 previously reported in those models and in patients with HD [6,19,21,22,25]. Consistent with our findings, previous HSF1 ChIP-seq analyses in ST*Hdh*^Q7^ and ST*Hdh*^Q111^ cells showed that mtHTT dramatically alters the genome-wide binding of HSF1, affecting genes associated with cytoskeletal binding, focal adhesion, and GTPase activity, several of which have synaptic functions [46]. We recently showed that rescuing levels of HSF1 in zQ175 mice not only resulted in increased *Dlg4* expression, but also ameliorated transcriptional alterations in signaling pathways related to synaptogenesis and glutamate receptor signaling and ameliorated many HD-like phenotypes [47]. Taken together, these data suggest that impaired HSF1 in HD largely has effects not only on PSD-95 regulation but also on other synaptic pathways. Further studies are needed to directly assess the association between synaptic transcriptional changes in HD and alterations in HSF1 genome-wide binding.

An important aspect of our study is the intersection between HSF1 regulatory function, *Dlg4* expression, and their connection with excitatory synapse density. We showed *Hsf1*^(+/−)^ mice resulted in a loss of thalamo-striatal (T-S) excitatory synapses, an important synaptic circuit involved in cognitive functions such as goal-directed learning, action selection, and flexible control of behavior, all of which are disrupted in HD [48,49]. Previous work showed working memory deficits in *Hsf1*^(−/−)^ mice [18] and a resemblance in several other behavioral abnormalities between *Hsf1*^(+/−)^ and *Hsf1*^(−/−)^ mice, including enhanced vulnerability to repeated stress exposure, reduced anxiety-like behavior, and altered locomotion activity [17]. Although cognition and memory has not been directly addressed in *Hsf1*^(+/−)^, based on the similarities in other behavioral abnormalities compared to *Hsf1*^(−/−)^ and the significant decrease in T-S synapse number we have observed, it is reasonable to predict that chronic depletion of HSF1 may cause memory-related deficits. Further studies to address this question are warranted.

The similarities between *Hsf1*^(+/−)^ and zQ175 mice regarding PSD-95 dysregulation and loss of T-S synapses further demonstrates the involvement of HSF1 in controlling excitatory synapse stability in the striatum. Reduction of T-S synapses precedes mtHTT aggregation and symptom onset in zQ175 mice and is believed to initiate striatal pathology [48,49]. Therefore, any therapeutic manipulations aimed at preventing T-S synapse loss could result in long-term benefits. Indeed, increasing the levels of HSF1 in zQ175 mice resulted in a rescue of T-S synapses [19] and increased the frequency of AMPA-mediated miniature excitatory postsynaptic currents at 12 months of age, which correlated with improved motor coordination [47]. Overall, these data provide a strong connection between HSF1, PSD-95, and the regulation of T-S synapse density.

In summary, our study provided evidence confirming the regulatory role of HSF1 in the direct regulation of synaptic components, such as PSD-95, and their connection with excitatory synaptic maintenance. Future analyses will be needed to assess the impact of HSF1 in the regulation of other synaptic genes whose expression is dysregulated in the context of aging and neurodegeneration, and their overall contribution to the pathological dysfunction in various synaptic circuits.

## 4. Materials and Methods

### 4.1. Cell Lines

Mammalian cell lines used in this study were the mouse-derived striatal cells ST*Hdh*^Q7/Q7^ and ST*Hdh*^Q111/Q111^ (Coriell Cell Repositories; Camden, NJ, USA). ST*Hdh*:*Hsf1*^+/−^ cells were generated using pSpCas9-(BB)-2A-GFP (PX458, Addgene #48138; Watertown, MA, USA) [50] and a gRNA-targeting HSF1- exon9 (5′-caccGAGTACCCGAGGGCTGTGAGGCTCATGGGCTCCCGACACTCCcaaa-3′). The efficacy of indel generation was tested using the SURVEYOR^TM^ nuclease assay. Transfected cells were sorted by FACS and individual GFP+ cells were plated into 96-well plates, generating a total of 10 plates per cell line. Each individual cell (clone) was assigned an ID based on the plate number and the position within the plate. Colonies from isolated clones were genotyped using HSF1 primers (Forward: 5′-CCTTAGTGGGTCAGCCTTTATG-3′, Reverse: 5′-AGGGGCATATCCCATTTCTAGT-3′). A total of 10 clones were preselected for each line based on genomic alterations on the *Hsf1* gene sequence and total HSF1 protein levels. PCR revealed a 248 bp deletion in ST*Hdh*^Q7^ clone 3A4, and a 321 bp deletion in ST*Hdh*^Q111^ clone 8E6, both of which presented a significant depletion in the levels of HSF1 protein compared to control cells. Cells were grown at 33 °C in Dulbecco’s modified Eagle’s medium (DMEM, Genesee; El Cajon, CA, USA) supplemented with 10% fetal bovine serum (FBS), 100 U mL^−1^ penicillin/streptomycin, and 100 µg mL^−1^ G418 (Gibco, Thermo Fisher Scientific; Waltham, MA, USA), as previously described [19].

### 4.2. Mouse Strains

For this study we used a full-length knock-in mouse model of HD known as zQ175 on the C57BL/6J background (The Jackson Laboratory Stock No. 027410; Bar Harbor, ME, USA) [30,51]. Sperm from HSF1 heterozygous knock-out (B6N(Cg)-Hsf1^tm1(KOMP)Vlcg^/JMmucd) mice was obtained from the Mutant Mouse Resource and Research Center (University of California, Davis; Davis, CA, USA) (Stock No. 048101-UCD) and generated by the Knockout Mouse Phenotyping Program (KOMP^2^). In vitro fertilization using C57BL/6N females was conducted at the Mouse Genetics laboratory at University of Minnesota. *Hsf1*^+/−: tm1^ mice were crossbred with CMV-CRE mice (B6.C-Tg(CMV-cre)1Cgn/J((Stock No. 006054) to delete the Neomycin cassette flanked by loxP sites (*Hsf1*^+/−: tm1.1:CRE^). The *Hsf1*^+/−: tm1.1:CRE^ line was crossed with C57BL/6N to remove the CRE gene (*Hsf1*^+/−: tm1.1^, referenced as *Hsf1* ^(+/−)^ in this study). All animal care and sacrifice procedures were approved by the University of Minnesota Institutional Animal Care and Use Committee (IACUC) in compliance with the National Institutes of Health guidelines for the care and use of laboratory animals under the approved animal protocol 2007-38316A.

### 4.3. Human Samples

HD brain tissues were obtained from Harvard Brain Tissue Resource Center (Belmont, MA, USA). Cases with and without clinical neurological disease were processed in the same way following the same sampling protocols. Control and HD cases were compared pairwise for sex and age.

### 4.4. Immunoblot Analysis

Sample preparation and immunoblotting conditions were performed as previously described [19]. Cell and striatum protein extracts from one hemisphere of mice were prepared in cell lysis buffer (25 mM Tris pH 7.4, 150 mM NaCl, 1 mM EDTA, 1% Triton-X100 and 0.1% SDS) supplemented with phosphatase and protease inhibitors (Thermo Scientific; Waltham, MA, USA). Protein samples were separated on 4–20% SDS Criterion TGX Stain-Free gels (BioRad; Hercules, CA, USA) and transferred to a nitrocellulose membrane (BioRad 0.2 µm). Primary antibodies used are as follows: anti-GAPDH (Santa Cruz, sc-365062, 1:10,000; Dallas, TX, USA), anti-HSF1 (Bethyl, A303-176A, 1:1000; Waltham, MA, USA), anti-PSD-95 (Novus, NB300-556, 1:1000; Centennial, CO, USA), and anti-Hsp70 (Enzo, C92F3A-5, 1:1000; Farmingdal, NY, USA). Quantitative analyses were performed using ImageJ software and normalized to GAPDH controls.

### 4.5. Immunohistochemistry & Synapse Density Analyses

Sample preparation was performed as previously described [19]. Fluorescent images from dorsal striatum (bregma 0.5–1.1 mm) were acquired on a confocal microscope (Olympus FV1000). Primary antibodies used are as follows: VGLUT2 (Millipore AB2251-I, 1:1000; Burlington, MA, USA), PSD-95 (Thermo Fisher 51-6900, 1:500; Waltham, MA, USA). Secondary antibodies used are as follows: goat anti-guinea pig Alexa 488 (1:200) and goat anti-rabbit Alexa 594 (1:200) (Invitrogen; Waltham, MA, USA). Confocal scans (optical section depth 0.34 mm, 15 sections per scan) in the dorsal striatum were performed at 60× magnification. Maximum projections of three consecutive optical sections were generated. Puncta analyses were conducted blinded using the PunctaAnalyzer Plugin (Durham, NC, USA) on ImageJ, as previously described [19,26]. For PSD-95 fluorescent intensity quantification, confocal scans in the dorsal striatum were performed at 60×. A minimum of three slices per animal and three animals per genotype and time point were analyzed. Intensity was normalized to the number of nuclei per scan section, and data was presented as % signal.

### 4.6. RNA Preparation and RT-qPCR

RNA was extracted from ST*Hdh* cells and mouse striatal tissues using the RNeasy extraction kit (Qiagen; Germantown, MD, USA) according to the manufacturer’s instructions. cDNA was prepared using the Superscript First Strand Synthesis System (Invitrogen; Waltham, MA, USA) according to the manufacturer’s instructions. SYBR green-based qPCR was performed with SYBR mix (Roche; Basel, Switzerland, & Genesee; El Cajon, CA, USA) using the LightCycler 480 System (Roche; Basel, Switzerland). Primers used are as follows: GAPDH (Forward: 5′ACACATTGGGGGTAGGAACA-3′, Reverse: 5′-AACTTTGGCATTGTGGAAGG-3′) and PSD-95 (Forward: 5′-CCGCGATTACCACTTTGTCT-3′, Reverse: 5′-ACGGATGAAGATGGCGATAG-3′). Each sample was tested in triplicate and normalized to GAPDH levels. For analysis, the ^ΔΔ^Ct method was used to calculate the relative fold gene expression.

### 4.7. siRNA Transfection

For HSF1 knock-down, ST*Hdh* cells were transfected at 75% confluency with a FlexiTube siRNA solution GS15499 (10 µmol from Qiagen; Germantown, MD, USA) using DharmaFECT1 transfection reagent as per the manufacturer’s instructions. As a negative control, non-targeting siRNA was used. Cells were incubated at 33 °C for 24 h. followed by RNA extraction and RT-qPCR. All siRNAs were validated by RT-qPCR and immunoblotting for knockdown efficiency.

### 4.8. Chromatin Immunoprecipitation

Chromatin immunoprecipitation in cell lines was performed as previously described [19]. Cells were grown to 75% confluency, placed on ice and cross-linked with 37% Formaldehyde followed by glycine quenching (125 mM). Chromatin shearing was performed using three sets of 40-s sonication at 20% amplitude. In the case of mouse tissue, 15 mg of frozen striatal tissue was crosslinked and quenched, as described for cell lines, and sonication was performed using three sets of 20-s pulses at 40% amplitude. In the case of human tissue, ~75 mg frozen tissue from caudate/putamen of postmortem samples obtained from Harvard biobank were crosslinked and quenched, as described for cell lines. Sonication of human tissue was performed using three sets of 20-s pulses at 50% amplitude. Input samples were saved prior to the addition of antibodies. Two milligrams of antibody (rabbit HSF1: Bethyl A303-176A; Waltham, MA, USA, and rabbit IgG: R&D Systems AF008; Minneapolis, MN, USA) was added to the cell and mouse samples and 4 µg antibody was used for human samples. Samples were incubated overnight at 4 °C. For cell lines, immunoprecipitation (IP) was performed using Protein G agarose beads (Invitrogen, 15-920-010; Waltham, MA, USA) and Dynabeads™ Protein G (Invitrogen, 10009D; Waltham, MA, USA) for mice and humans, followed by chromatin purification using the Qiaquick min-elute PCR purification kit (Qiagen; Germantown, MD, USA) per the manufacturer’s instructions. SYBR green (Roche; Basel, Switzerland, and Genesee; El Cajon, CA, USA) qPCR was performed on IP and input samples. ^ΔΔ^Ct method was used to determine the relative amounts of DNA normalized to input. Binding of HSF1 was evaluated using primers spanning the four different HSE on the murine and human PSD-95 promoter/gene (HSE 1#; Forward:5′-GAGCCACAAACAGTCGAA-3′, Reverse: 5′-TGGAAAGTGGCAGATGAGTG-3′; HSE #2: Forward: 5′-CCCACCTCTCCTAGCACAT-3′, Reverse: 5′-ATCATGAGACCC-3′; HSE #3: Forward: 5′-GGTCTTTGAGGGGGTGATCT-3′, Reverse: 5′-CTGACCTGGGAGCTGGTAAA-3′; HSE #4: Forward: 5′-TCTCCTCCTCTCTCCCCTTC-3′, Reverse: 5′-CACACCCCGATTCTCAGG-3′); and human HSE: Forward: 5′-TCACTGCCCCTCCCTTAGTA-3′, Reverse: 5′-GGGGTTTTACGGGTAAGAGG-3′), and values were normalized against IgG.

### 4.9. Luciferase Assays

We used the Dual-Luciferase^®^ Reporter (DLR™) Assay (Promega, E1910; Madison, WI, USA) per manufacturer’s instructions. We used PSD95-pGL3a (a generous gift from Dr. Bao, Washington University, St. Louis, MO, USA) [8] and substituted the firefly luciferase gene with the pGL4-derived firefly. Mutant HSE PSD95-pGL3a was generated by site-directed mutagenesis to substitute GA→CT within the HSE. Two milligrams of modified PSD95-pGL3a or mut. HSE PSD95-pGL3 and 2 µg of Renilla luciferase vector (control reporter) were co-transfected into STHdh cells and incubated for 12 h. Cells were then harvested and plated into a 96-well plate to measure luciferase activity. Luciferase activity (luminescence) was calculated by dividing the signal from Firefly and Renilla luciferase and was relativized to the control sample.

## Figures and Tables

**Figure 1 ijms-22-13113-f001:**
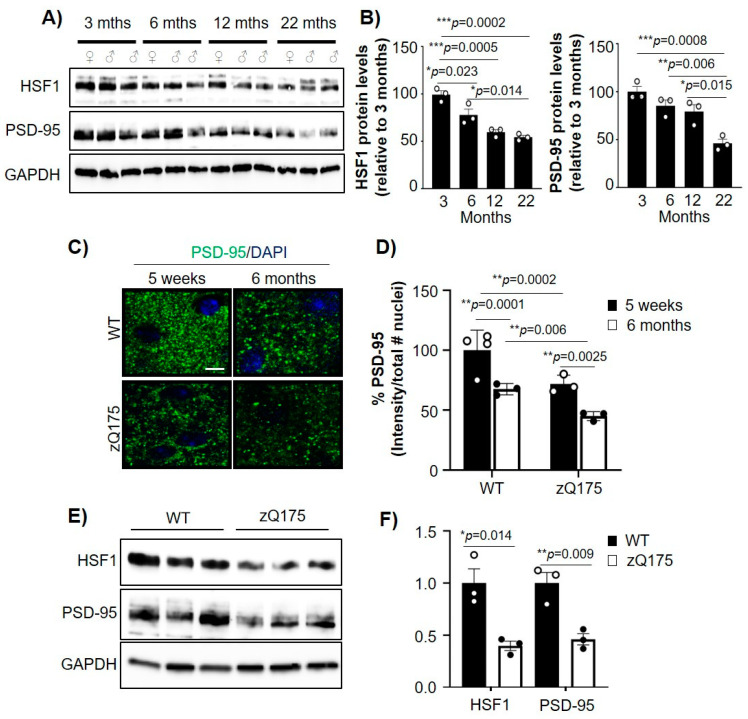
mtHTT exacerbates age-dependent down-regulation of HSF1 and PSD-95. (**A**) Immunoblot for HSF1 and PSD-95 from striatum samples from WT mice at different ages. GAPDH is used as a loading control. (**B**) HSF1 and PSD-95 protein levels quantified from A using ImageJ analyses. Data was normalized to GAPDH levels and relativized to 3 months (*n* = 3 mice/genotype). (**C**) PSD-95 immunofluorescence in the striatum of WT and zQ175 mice. Scale bar 5 µm. (**D**) PSD-95 intensity quantified by ImageJ from images in (**C**). Data was normalized to number of nuclei, relativized to 5 weeks WT, and shown as % PSD-95 signal (*n* = 3–4 mice/genotype). (**E**) Immunoblotting for WT and zQ175 at 6 months (*n* = 3 mice/genotype). GAPDH was used as a loading control. (**F**) HSF1 and PSD-95 protein levels quantified from (**E**) using ImageJ. Data was normalized to GAPDH levels and relativized to WT. Error bars denote mean ± SEM. One-way ANOVA with Tukey’s post-hoc correction in (**B**), two-way ANOVA with Tukey’s post-hoc correction in (**D**), and unpaired Student’s *t*-test in F. * *p* < 0.05, ** *p* < 0.01, *** *p* < 0.001. Only significant *p*-values are shown. Uncropped blots can be found in Appendix A.

**Figure 2 ijms-22-13113-f002:**
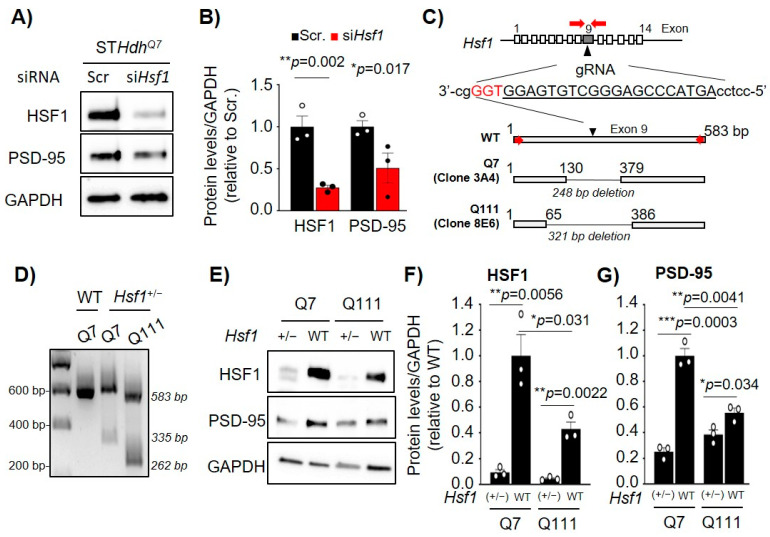
Acute or chronic reduction of HSF1 levels decreased PSD-95 levels in immortalized striatal cells. (**A**) Immunoblot of ST*Hdh*^Q7^ cells transfected with non-targeting siRNA (Scr.) or si*Hsf1*. GAPDH is used as a loading control. (**B**) HSF1 and PSD-95 protein levels quantified from (**A**) using ImageJ analyses. Data was normalized to GAPDH levels and relativized to Scr (*n* = 3), Scr (black), and *siHsf1* (red). (**C**) Genetic diagram representing the generation *of Hsf1* heterozygous (+/−) ST*Hdh* cell lines using CRISPR/Cas9. Guide RNA (gRNA) targets position +174 within exon 9 of *Hsf1*. Cas9 cleavage generated a 248bp deletion in ST*Hdh^Q7^* (clone 3A4) and a 321bp deletion in ST*Hdh*^Q111^ (clone 8E6). (**D**) PCR analyses from genomic DNA obtained from WT, clone 3A4, and clone 8E6 *for Hsf1* exon 9 DNA region. (**E**) Immunoblotting for WT and ST*Hdh* cell lines lacking one allele of *Hsf1*. GAPDH was used as a loading control. (**F**) HSF1 and (**G**) PSD-95 protein levels quantified from (**A**) using ImageJ analyses. Data was normalized to GAPDH levels and relativized to WT (*n* = 3). Error bars denote mean ± SEM. One-way ANOVA with Fisher’s LSD correction in (**B**), unpaired Student’s *t*-test in (**F**), (**G**). * *p* < 0.05, ** *p* < 0.01, *** *p* < 0.001. Uncropped blots can be found in Appendix A.

**Figure 3 ijms-22-13113-f003:**
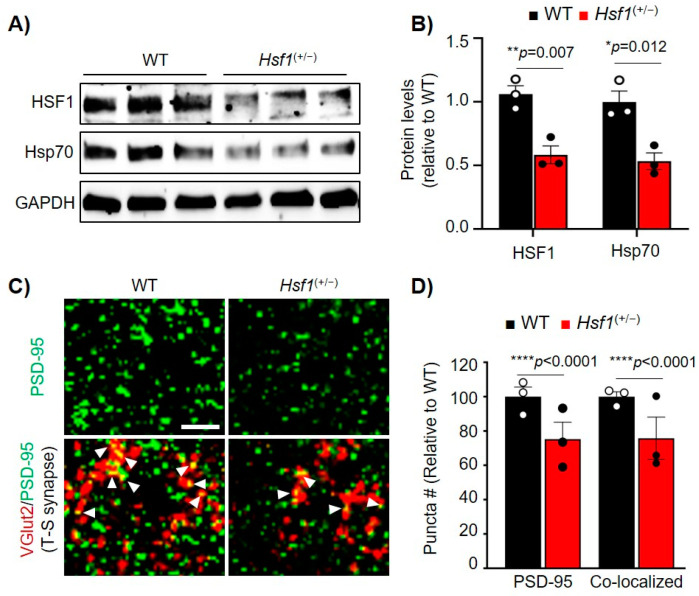
HSF1 haploinsufficiency decreased PSD-95 puncta and striatal synapse density. (**A**) Immunoblot from striatum samples of WT (*Hsf1* ^(+/+)^) and *Hsf1* ^(+/^^−)^ mice at 2.5 months. GAPDH is used as a loading control. (**B**) HSF1 and Hsp70 Protein levels quantified from (**A**) using ImageJ analyses. Data was normalized to GAPDH levels and relativized to WT. (**C**) Immunohistochemistry for PSD-95 (green) and the pre-synaptic vesicular protein VGlut2 (red) in the striatum of WT and *Hsf1*^(+/^^−)^ at 12 months. Arrows indicate co-localization between PSD-95 and VGlut2. Scale bar 5 µm. (**D**) Puncta analysis for PSD-95 and co-localized PSD-95/VGlut2 from (**C**). Punta number (#) was relativized to WT. Error bars denote mean ± SEM. *n*=3 mice/genotype. Unpaired Student’s *t*-test. * *p* < 0.05, ** *p* < 0.01, **** *p* < 0.0001. Uncropped blots can be found in Appendix A.

**Figure 4 ijms-22-13113-f004:**
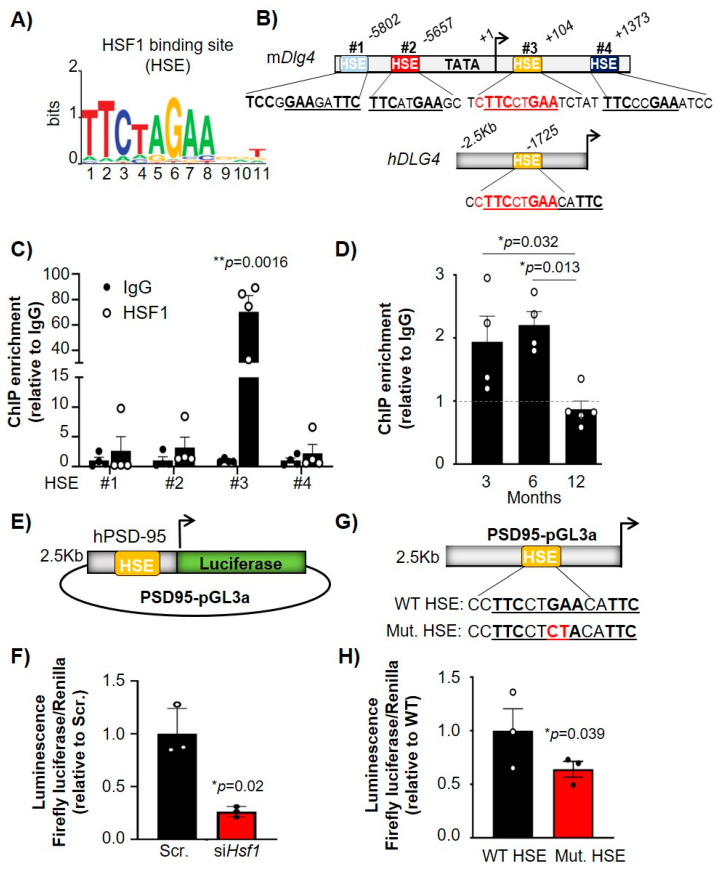
HSF1 binds to HSE within the mouse and human Dlg4 (PSD-95) promoter/gene and directly regulates Dlg4 transcription. (**A**) HSF1-binding motif (HSE) represented by MotifExpress. (**B**) Genetic diagram of the HSEs found in the promoter and intergenic region of PSD-95 identified in silico. Mouse *Dlg4* (mPSD-95) and human *DLG4* promoter (h*DLG4*) share an identical HSE (yellow). All HSE sequences are shown in 5′-3′ as they appear in the sense (top) strand. Note that HSE #1 motif is present in the antisense (bottom) strand and represents an inverted HSE. (**C**) HSF1-ChIP for the four different HSEs identified in the m*Dlg4* gene. Data is relative to IgG (*n* = 4). (**D**) HSF1-ChIP for HSE #3 in the striatum of WT mice at 3 (*n* = 4), 6 (*n* = 4) and 12 months (*n* = 6). (**E**) Diagram of PSD95-pGL3a vector expressing Luciferase under control of h*DLG4* promoter. (**F**) Luciferase activity in ST*Hdh*^Q7^ cells transfected with non-targeting siRNA (Scr.) or si*Hsf1*. Data is normalized to Renilla luciferase and relativized to Scr. (**G**) Diagram of PSD95-pGL3a containing a WT HSE or a mutant HSE (Mut. HSE). (**H**) Luciferase activity in ST*Hdh*^Q7^ cells transfected with PSD95-pGL3a containing WT or Mut. HSE. Data is normalized to Renilla luciferase and relativized to WT. Error bars denote mean ± SEM. Unpaired Student’s *t*-test in (**C**,**F**,**H**) and One-way ANOVA with Tukey’s post-hoc test in (**D**). * *p* < 0.05, ** *p* < 0.01.

**Figure 5 ijms-22-13113-f005:**
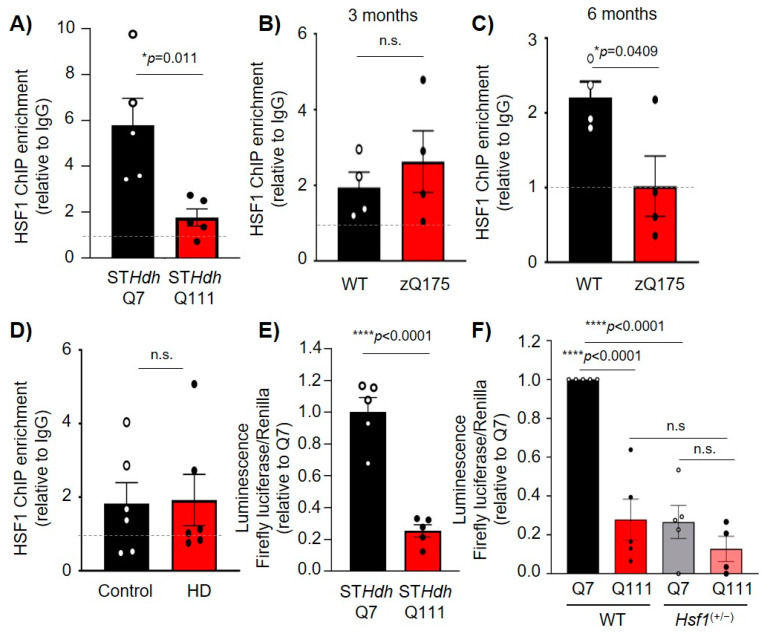
HSF1 binding to regulatory elements of PSD-95 and its regulation is impaired in HD. (**A**–**D**) HSF1-ChIP on PSD-95 HSE #3 in (**A**) ST*Hdh*^Q7^ and ST*Hdh*^Q111^ cells (*n* = 6), (**B**) the striatum of WT and zQ175 mice at 3 (*n* = 4 mice/genotype) and (**C**) 6 months (*n* = 4 mice/genotype), and (**D**) in the striatum of patients with HD and sex and age-matched controls (*n* = 6 individuals/group). (**E**) Luciferase activity in ST*Hdh*^Q7^ and ST*Hdh*^Q111^ cells. Data is normalized to Renilla luciferase and relativized to Q7 cells (*n* = 5). (**F**) Luciferase activity in WT and *Hsf1*^(+/−)^ ST*Hdh*^Q7^ and ST*Hdh*^Q111^ cells. Data is normalized to Renilla luciferase and relativized to Q7. Error bars denote mean ± SEM. Unpaired Student’s *t*-test in (**A**–**E**), and One-way ANOVA with Tukey’s post-hoc test in (**F**). * *p* < 0.05, **** *p* < 0.0001.

## Data Availability

All data generated in this study is presented in the current manuscript. No new datasets were generated. Data is available upon request from the corresponding author.

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
