# Peer review of "Heat Shock Factor 1 Directly Regulates Postsynaptic Scaffolding PSD-95 in Aging and Huntington’s Disease and Influences Striatal Synaptic Density"

_ijms, 2021, doi:10.3390/ijms222313113_

Round 1
Reviewer 1 Report
In this study, Zarate et al. present evidence supporting a direct role of HSF1 in the regulation of PSD-95. They report that HSF1 reduction, either by HSF RNA silencing in cell lines or in the heterozygous mouse model HSF1, decreases PSD-95 expression. In this mouse model, the authors also describe a significant reduction of PSD-95 puncta. The authors identify the binding of HSF1 through HSE sequences within the mouse and human Dlg4 gene, and they demonstrate how this interaction regulates directly PSD-95 expression through luminescence activity assays. Finally, the authors show that this binding is impaired in cell and mouse models of HD.
In general, the experiments are properly executed and their interpretation is consistent with the data. However, there are a number of weaknesses in the manuscript, and I have various comments and suggestions, detailed below.
Major points:
- Figure 1 and paragraph 2.1 seem to recapitulate already published information regarding HSF1 and PSD95 decrease over age and in zQ175 mice (REF 4, 19, 25 and 35). Although the authors present a longitudinal study of such changes over time, this does not bring sufficient novelty to be shown as a main figure. The authors might discuss this along the text and move Figure 1 (panels A,B,F,G and H) to Supplementary materials. Panels C, D and E can be omitted since no new information is shown (REF 4, 19, 25 and 35).
- The title of section 2.1 in the Results is not accurate enough: the authors do not directly demonstrate that mHTT exacerbates the depletion of PSD-95 and HSF1, they just show a depletion in HD cells, mouse models and post-mortem tissue. Again, in the Discussion (line 356), this should be changed.
- In figure 2D, why the Q7 (Hsf1+/-) WT band does not migrate at the same level as the other ones?
- The title of Figure 3 legends does not correspond to what the authors are showing, there is a decreased signal in PSD-95 puncta but not a decrease in protein levels. In order to address the protein levels they should perform a Western blot. In addition, the sentence “In analyzing PSD-95 puncta by IF instead of total PSD-95 protein levels by immunoblotting, we were able to assess only the levels of synaptic PSD-95 in mature spines without considering non-synaptic, dendritic shaft levels of PSD-95” is not clear: do the authors want to indicate that this is the only accurate way to address this point? It would be interesting to analyse total PSD-95 protein levels by immunoblotting and show it in Supplementary materials. Again, in the Discussion (line 376) this should be re-written.
- In Figure 5D), it would be more appropriate if the authors represent a fold enrichment respect to IgG as they did in other panels (Figures 4C, 5A, B and C). The decrease seen in HSF1 and IgG in control vs HD is similar; therefore, it is probable that no changes will be detected if represented with fold enrichment. Furthermore, as IgG is a control and it shows more tendency than HSF1 to decrease in HD, the authors should not suggest that there is a reduction in HD.
- The description in the text of Figure 5F is misleading as there is no statistically significant reduction between Q111 Hsf1+/+ and Q111 Hsf1+/- in luciferase activity (line 279 to 282). In fact, it seems that PSD95-pGL3a luminescence reduction in Q111 cells is independent of Hsf1 partial KO.
- In line 283, the sentence: “Taken together our data indicated the pathological depletion of HSF1 in HD results in decreased HSF1 binding to PSD-95 and subsequent decreased expression of this synaptic protein” may be changed to “Taken together our data indicated the pathological depletion of HSF1 in HD cell lines and mouse models results in decreased HSF1 binding to PSD-95 and subsequent decreased expression of this synaptic protein”, as in HD patients it does not reach statistical significance.
- When the authors discuss the difference over the time in the reduction of HSF1 and PSD-95 (HSF1 reduction starts at 3 months and PSD-95 reduction starts at 12 months), they state: “Alternatively, the stabilization of PSD-95 protein can be regulated by post-translational events [49] even when the mRNA levels are depleted, explaining how the protein is maintained for a period of time before its eventual depletion”. However, it sounds improbable to maintain protein levels via post-translational mechanisms across several months. Thus, this sentence may be omitted or further discussed.
- In general, the Discussion section is quite long, and it is redundant to some extent with the Introduction section. The authors may omit repeated references and ideas to shorten this section.
Minor points:
In Figure 1:
- In A) the bands presented in this panel seem to be enlarged from the original ones in Supplementary material.
- In B) HSF1 protein levels are relative to 3 months, not to WT.
- In C) HSF1 protein levels are relative to 3 months, not to WT.
- In D) relative units, such as percentages relative to 5 weeks old WT, would be easier to read. Also, the bars of p-values are displaced.
- In C) and D) please unify the name of the genotype (use zQ175 or KIQ175).
- In E), the authors must show histograms with individual points, error bars and statistical tests in order to assume that HSF1 and PSD-95 levels are diminished. In the legends, “n=3” is indicated but this is not shown in the uncropped blots.
- In H) please indicate that the histograms correspond to human mRNA levels.
- In the figure legends, there is no description of G) panel. The description after G) seems to correspond to H) panel.
In Figure 2:
- In B) the protein levels are relative to Scr, not to WT, and there is a lack of legends indicating that colour black corresponds to Scr and red to siHsf1. In the legends, please indicate that levels where normalized to scramble. Individual points in the histograms would be useful.
- In E) the superindex lettering of the genotypes are difficult to read.
- In F) the protein levels are relative to Hsf1+/+, not to WT (although Hsf1+/+ is equivalent to WT, the authors should unify the names).
- There is a lack of G) lettering in the figure and the legends.
- In F) and G) individual points in the histograms would be useful.
- At the end of the figure legends, the last sentence is repeated twice.
In Figure 3:
- The authors should choose between Hsf1+/+ or WT, using both denominations is misleading.
- In A) it seems that the bands presented in this panel do not correspond to the ones shown in uncropped blots.
- In D) please substitute the “y” axis subtitle by “(relative to WT)”.
In Figure 4:
- It seems that HSE #1 in B) is incorrectly spelled, as it does not correspond to the motif presented in A).
- In A) when using MotifExpress, the authors should reference the tool in the materials and methods section.
- Please change the title of the legends and letter H) to bold type.
- In line 245, mPSD95 gene should be written in italics. It would be advisable to use the same denomination as for the human gene, i.e mDlg4.
- In general, it would be useful if the authors provide individual points in the histograms.
In Figure 5:
- In the legends, (D) must be at the end of line 269.
- In general, it would be useful if the authors provide individual points in the histograms.
- Panel F) does not have any legend.
- The references to panels G) and H) do not correspond to any panel.
The authors should be consistent in the nomenclature of Dlg4 gene vs PSD-95 protein, and when referring to the mouse gene it should be written in italics (lines 213/214/229), as well as when referring to HSEs (line 221).
In materials and methods:
- In Mouse strains, please reference zQ175 mice generation (Heikkinen 2012 or Menalled 2012, for example).
- Every micro (µ) symbol is wrong, the authors must write it correctly.
In the text (line 165) the reference to Supplementary Figure 1 is not correct.
In the text (line 206) it would be more appropriate to use the term “reduction” instead of “depletion” (and all along the manuscript).

Reviewer 2 Report
This is a well written manuscript providing novel information on the regulation of PSD-95 through HSF1 in an age and disease-dependent manner. Please see below some moderate and numerous minor concerns which once addressed would strengthen this manuscript significantly.
The authors provide evidence that HSF1 and PSD-95 exhibit age-dependent protein loss in WT mouse striatum (Fig. 1 A+B). The reviewer is wondering whether other brain regions show equal changes in these proteins, or whether this observation is specific to the striatum. This would also be important to examine for the co-localization studies described in Figure 3.
What happens to other synaptic proteins (pre and postsynaptic)? The authors could have easily run additional western blots of PSD-95 family members, such as SAP97, or pre-synaptic proteins such as synapsin 1 or synaptophysin – all of which were mentioned in the discussion to be regulated by HSF1. This seems like a missed opportunity.
The authors state that in HD, the loss of HSF1-regulation of PSD-95 is exacerbated. The reviewer is not quite convinced of this statement. To truly state that the decrease in HSF1 and PSD-95 is exacerbated in disease, the samples need to be run on the same gel to directly compare (see more comments on Figure 1 below). Similarly, in Figure 5 F - the loss of Hsf1 in Q111 cells does not significantly reduce the promoter activation of PSD-95 suggesting that maybe there is no disease-associate exacerbation of HSF1-dependent PSD-95 regulation?
In Figure 2, the authors try to connect the loss of HSF1 to loss of PSD-95. It might be helpful to actual analyze the data as a ratio between HSF1 and PSD-95, which would take into account that in Q111 cells, PSD-95 is already reduced (panel 2F), hence one would need to figure out the relative reduction of PSD-95 upon siRNA treatment for hsf1.
The authors use hsf1 +/- mice to show decreased PSD-95 and decreased synapse numbers as shown by co-localization with VGlut2. It would be great to provide some input on the behavioral aspects of these mice – did the authors look at behavior at all in these animals? Do they behave normally given the loss of PSD-95 and haploinsufficiency of HSF1? How do they compare to the full KO mice, which the authors describe in the text showing working memory deficits? This information could be helpful in the context of overall relevance of the loss of synapses described.
Minor comments
The reviewer would prefer if the authors showed all their data using dot blots, which will allow the reviewer to see the spread of the individual data sets and the sample numbers more easily.
Figure 1 G/H figure legend does not match the panels in the figures.
Figure 1B – while the stats for HSF1 seems to be referred to the 3 months time point, the stats for PSD-95 are referred to the 22 months time point, while they should equally be shown in reference to the 3 months time point, just like HSF1.
Figure 1C/D/E – to truly compare protein levels between WT and zQ175 mice, these western blots should be run on the same gel.
Figure 1D – the y-axes should be labeled differently, but most importantly, the reviewer is unsure why the authors decided to show intensity of PSD-95 per total nuclei? Clearly the microscopic image will contain dendrites from neurons whose cell body is not within the microscopic view. Furthermore, how many brain sections per mouse were imaged for these quantifications? Finally, the x-axes is labeled with WT and ‘KIQ175’ – is KIQ175 the same as zQ175?
Figure 2
While the figure does not show panel G, the manuscript text refers to Figure 2G
The authors state the two clones were selected from their CRISPR-Cas9 Hsf1 knockdown – but not details were provided based on what these two clones were picked.
Figure 3
For co-localization quantifications, using a Pearson’s coefficient is the most appropriate quantitative analyses and should be applied to these images
Figure 5
The figure panels do not match the figure legends E/F vs G/H?
